# Recent Progress in Double-Layer Honeycomb Structure: A New Type of Two-Dimensional Material

**DOI:** 10.3390/ma15217715

**Published:** 2022-11-02

**Authors:** Ming-Yu Ma, Dong Han, Nian-Ke Chen, Dan Wang, Xian-Bin Li

**Affiliations:** 1State Key Laboratory of Integrated Optoelectronics, College of Electronic Science and Engineering, Jilin University, Changchun 130012, China; 2State Key Laboratory of Luminescence and Applications, Changchun Institute of Optics Fine Mechanics and Physics, Chinese Academy of Sciences, Changchun 130033, China; 3Department of Mechanical Engineering and Materials Science, Yale University, New Haven, CT 06511, USA

**Keywords:** two-dimensional materials, double-layer honeycomb, nanostructure, material design, optoelectronic semiconductors

## Abstract

Two-dimensional (2D) materials are no doubt the most widely studied nanomaterials in the past decade. Most recently, a new type of 2D material named the double-layer honeycomb (DLHC) structure opened a door to achieving a series of 2D materials from traditional semiconductors. However, as a newly developed material, there still lacks a timely understanding of its structure, property, applications, and underlying mechanisms. In this review, we discuss the structural stability and experimental validation of this 2D material, and systematically summarize the properties and applications including the electronic structures, topological properties, optical properties, defect engineering, and heterojunctions. It was concluded that the DLHC can be a universal configuration applying to III–V, II–VI, and I–VII semiconductors. Moreover, these DLHC materials indeed have exotic properties such as being excitonic/topological insulators. The successful fabrication of DLHC materials further demonstrates it is a promising topic. Finally, we summarize several issues to be addressed in the future, including further experimental validation, defect engineering, heterojunction engineering, and strain engineering. We hope this review can help the community to better understand the DLHC materials timely and inspire their applications in the future.

## 1. Introduction

Two-dimensional (2D) materials have come to be a big family with various categories including elemental 2D materials, transition metal dichalcogenides, MXenes, metals, metal-organic frameworks, and so on [1,2,3,4]. The special structural features and physicochemical properties make them one of the most attractive candidates for a broad variety of potential applications, such as high-mobility transistors, high-performance data storage/computing applications, and photovoltaic applications [5,6,7,8,9,10]. Finding new kinds of 2D materials often provides opportunities for new physics or applications and thus is an important topic in the 2D-material community. Generally, 2D materials are most easily obtained from van der Waals (vdW) layered materials, such as graphene, transition-metal dichalcogenide, and black phosphorus [9,10,11,12,13,14,15]. This is because the weak vdW interactions between the layer blocks can easily be broken and thereby corresponding 2D counterpart can be achieved by physical or chemical exfoliations [16]. Meanwhile, it is also attractive to achieve 2D materials from traditional bulk materials without vdW interactions because it not only expands their categories but also provides a new paradigm for designing 2D materials.

In fact, efforts have been carried out to achieve 2D materials from bulk materials. A key question is: what are the stable structures when 3D materials are reduced to their 2D limits? Graphene exfoliated from graphite has a well-known planar honeycomb lattice [17]. The stability of graphene benefits from the strong sp^2^-hybridized bonds as well as π bonds between carbon atoms [18]. However, such a configuration may be not stable enough for other compounds. When the atomic radius or distances are larger than that of carbon, the π-bonds are weaker, and the bonding type turns from sp^2^ towards sp^3^-like, which can make the planar honeycomb structure form a buckled honeycomb structure. Many 2D materials with such a buckled honeycomb have been found or predicted including silicene [19,20], germanene [21,22], stanene [23,24,25], III–V compounds including InP, InAs, InSb, GaAs, AlSb [26,27,28], II–VI compounds including IIB selenides and tellurides [29].

However, the configuration of a single buckled layer honeycomb (SLHC) is still not stable enough and often fails for solids like the group I–VII semiconductors in their 2D limit. Therefore, new configurations that can stabilize traditional semiconductors in their 2D limit are still required to be investigated. In 2018, Lucking et al. theoretically propose a 2D structure named double-layer honeycomb (DLHC) [30], which is demonstrated to have lower formation energy than the truncated bulk and other 2D counterparts, such as in CdTe [31,32], AlSb [33,34], and InAs [35]. Later, several such kinds of 2D materials are successfully fabricated in experiments [34,36,37,38]. The configuration of DLHC offers a new way to achieve 2D materials from traditional 3D semiconductors including groups III–V, II–VI, and I–VII compounds. Especially, the I–VII 2D materials are rarely realized by other configurations before [30]. Therefore, the DLHC materials not only expand the types of 2D materials but also provide an opportunity of designing new physical/chemical properties and even new applications in the 2D limit.

In this review, to help the community better and timely understand the DLHC structure and to draw attention to its further development, we introduce the structure, properties, potential applications, and underlying mechanisms of the DLHC 2D materials. Firstly, the structure, stability, and corresponding bonding mechanism of DLHC are presented. Then, recent progress in experimental fabrications of DLHC materials is introduced. Next, the properties, including the band structures, the topological properties, optical properties, chemical properties, and defect engineering of DLHC materials are discussed in detail. Finally, the multilayer and heterojunction of DLHC materials are also introduced briefly. Outlook related to DLHC materials is also presented. We hope this review can be helpful to understand the DLHC materials better and provide guidance to accelerate the development of this rising DLHC nanomaterial for future applications. 

## 2. Discussion

### 2.1. Theoretical Prediction

To clearly illustrate the atomic structure of DLHC, the construction of DLHC from a traditional zinc blende bulk is shown in Figure 1a,b following Lucking’s work of Ref. [30] in 2018. Firstly, the zinc blende bulk is truncated by the (111) plane to get a double-layer slab which can be regarded as two SLHC layers (see Figure 1a). The truncated bulk is not stable due to the existence of dangling bonds on the surfaces. Then, the outer-layer cations are moved to the site directly above the anions (in the lower SLHC) as indicated by the red arrows in Figure 1a. In this way, the stable DLHC structure is built as shown in Figure 1b. Indeed, the calculated total energy of DLHC is lower than that of the slab truncated from its bulk counterparts.

The electronic origin of the stability is attributed to the elimination of dangling bonds by forming new bonding interactions. Figure 1c,d shows the electron localization functions (ELF) of the truncated bulk and DLHC, respectively. Compared to the truncated bulk, the number of interlayer bonding (between two SLHC layers) in DLHC is doubled while the bonding type becomes more ionic. The interlayer bonding can also be characterized by the charge density difference (CDD) between DLHC and the two isolated SLHCs as shown in Figure 1e. By forming the DLHC structure, electrons are transferred from Ga atoms of two SLHCs to interlayer bonds.

The bonding mechanism implies that DLHC can be, to some extent, a general way to achieve 2D materials from traditional zinc-blende semiconductors. Figure 1g shows the calculated formation energies of DLHCs with different compositions. Most of the DLHCs are demonstrated to be stable by calculated phonon spectra [30]. Figure 1h–k shows several examples of the phonon spectra of DLHCs, including the III–V group (such as GaAs and AlSb), II–VI group (such as ZnTe), and I–VII group (such as AgI). The relatively small formation energies also suggest the possibility of experimental fabrications. Finally, the stability of DLHC GaAs is further verified by the molecular dynamics (MD) simulations at 600 K (see Figure 1f). Their band structures mainly show the typical electronic structure of semiconductors, which determines the electrical and optical properties of materials. Most II–VI and I–VII DLHCs have band gaps larger than their bulk materials, which is consistent with a stronger quantum confinement effect at their 2D limit. However, most III–V DLHCs have smaller band gaps than bulk. A possible reason is that the bonding type transforms from covalent to ionic when the bulk is turned to its DLHC counterpart. Further explorations are still needed.

### 2.2. Experimental Fabrication

Several 2D DLHC materials indeed have been experimentally fabricated most recently. Not only the atomic configuration but also the electronic structure is consistent with theoretical calculations. For example, in 2021, Qin et al. fabricated monolayer DLHC of AlSb on graphene-covered single-crystal SiC (0001) by molecular beam epitaxy (MBE) [34]. The vdW interaction of graphene is important for the direct growth of DLHC AlSb because the DLHC AlSb failed to be grown on a Si substrate. Using an optimal substrate temperature (170 °C), AlSb films with a uniform height of 0.844 nm are realized (Figure 2a–c). The theoretical height of DLHC AlSb is about 0.396 nm. Considering the vdW gap between DLHC AlSb and graphene, the total theoretical height of a monolayer DLHC AlSb on graphene is about 0.767 nm which is close to the measured value. Moreover, the spectroscopic imaging scanning tunneling microscopy (STM) image (Figure 2d) exhibits triangular lattices (Figure 2e,j) with a lattice constant of 0.42 nm that agrees well with the theoretical lattice constant of 0.429 nm. Therefore, the AlSb film in Figure 2b can be identified as monolayer DLHC AlSb. Figure 2f shows a cross-sectional imaging scanning transmission electron microscopy (STEM) image of the DLHC AlSb capped with a protecting Sb layer where the height of DLHC AlSb is determined to be 0.76 nm which is in excellent agreement with the theoretical value.

Then, the electronic structure of DLHC AlSb is experimentally measured and compared with that of theoretical calculations (Figure 2g–l). Figure 2g shows the tunneling conductance spectrum corresponding to the density of states (DOS). The profile agrees well with the calculated DOS in Figure 2h. Moreover, the experimental band structures along Γ-K (Figure 2k) and Γ-M (Figure 2l) directions also coincide with theoretical band structures (Figure 2i) for the following two reasons: (1) they both show a direct band gap of 0.93 eV; (2) the conduction bands are narrower than the valence bands, while among them the dispersion along the Γ-M direction is more inflated than that along Γ-K direction.

As mentioned above, I–VII semiconductors are hard to be grown as 2D materials compared with IV, III–V, and II–VI semiconductors. Therefore, the successful fabrication of the I–VII DLHC in experiments demonstrates that the DLHC indeed is a reliable way to get new 2D materials. In addition, I–VII DLHCs have been proven to be excellent direct band gap semiconductors with thermodynamic stability and high carrier mobility [39]. In fact, 2D DLHC has been regarded as a kind of sandwich-like structure as early as 2009 [36]. Figure 3a,b shows the side and top views of the sandwich-like model of the AgI ultrathin film on the (001) surface of the Ag substrate with an iodine buffer layer between them. Comparing the atomic model with that in Figure 1b, it can be found that the sandwich-like thin film is a distorted DLHC structure. The STM images (Figure 3c) demonstrate a visible superstructure with a periodicity quasi-hexagonal atomic modulation. The minimum step height of the STM image is approximately 7.5 Å in line with the height of the DLHC on a vdW surface. In contrast, a study in 2022 [37] reports an undistorted DLHC AgI grown on the (111) surface of Ag substrate with an iodine buffer layer between them (Figure 3d–f). A clearly visible hexagonal superstructure with a quasi-hexagonal atomic modulation can also be observed in the STM images (Figure 3f), which is also in keeping with the simulated STM image of DLHC.

In another work in 2022, both the DLHC AgI and the DLHC CuI sandwiched by two graphene layers are fabricated under room temperature [38]. The DLHC AgI here is also undistorted with a suggestion that it may be the most stable DLHC configuration on vdW surfaces. The theoretical model and experimental image of DLHC CuI are shown in Figure 3g–i. It seems interesting that the DLHC CuI without the encapsulation of graphene is not stable at room temperature but it is stable at high temperatures between 645 and 675 K [38]. The lattice mismatch between graphene and DLHC CuI leads to a 1.6% compression strain in DLHC CuI. First-principles calculations using the HSE06 hybrid functional suggest DLHC CuI has a direct band gap of 3.17 eV.

Table 1 compares the different synthesis conditions which should be helpful to guide the fabrication of more DLHC materials. The inert buffer layer modulates the stability of the DLHC, allowing them to grow easily and even at room temperature under the weak vdW interaction. These synthetic procedures provide access to other unsynthesized DLHC in the experiment.

### 2.3. Excitonic Insulator

An excitonic insulator (EI) refers to a material whose exciton binding energy is larger than the band gap. Such an exotic phenomenon is not easy to be observed in traditional bulk semiconductors due to quite a small exciton binding energies [40]. 2D materials are more likely to form EI because (1) they often have greater exciton binding energy than 3D materials on account of the weak screening that strengthens the electron-hole interaction; (2) the band gap can be more effectively reduced by electric fields or strain [41], which means the binding energy (E_b_) and band gap (E_g_) can be decoupled. Excitingly, DLHC GaAs have been predicted to be a good platform to study EI [30,42]. Figure 4a shows the schematic picture of the indirect and direct gap excitonic instability, where holes and electrons form excitons through a mutual Coulomb attraction. Figure 4b shows the atomic structure of DLHC GaAs as mentioned before. Its valence band maximum (VBM) is above the conduction band minimum (CBM) as shown in Figure 5b and the band-edge states have the same parity which prohibits the electronic transition and thus can stabilize the EI [30]. On the other hand, the band gap of DLHC GaAs can be effectively tuned by in-plane strains. Tensile strains turn it into a metallic state while compression gradually opens the band gap (see Figure 4c). In contrast, the E_b_ is almost invariable under different strains. Figure 4d summarizes the change in E_g_ and E_b_ with the change of strain. It reveals the DLHC GaAs is a spontaneous EI with E_b_ > E_g_ under the strain range of −2% to 1% calculated by HSE. Therefore, it can be expected that there exist other DLHC EI materials and further investigations are desired.

### 2.4. Topological Insulator

In recent years, there have been many studies on topological insulators, and one study reported the high-throughput screening of 13 topological insulators from 1825 monolayers, focusing on easily/potentially exfoliated 2D materials with good properties in low-dissipation nanowires and topological field-effect transistors [43]. Topological insulator properties are worth being investigated in DLHC materials. The interesting discovery about DLHC is that as the number of layers increases, the parity of the band edges and the corresponding property of the topological insulator would change. Also taking DLHC GaAs as an example, the 2-layer DLHC of GaAs can become a topological insulator [30], while the DLHC is not. Figure 5a,b shows the band structures of SLHC and DLHC, respectively. The VBM is a bonding state and the CBM is an antibonding state for SLHC while the VBM and CBM are both antibonding states for DLHC. For the monolayer DLHC, the inversion between two states with the same parity has not affected the topological properties, that is the topological invariant Z_2_ = 0 (Figure 5c). When it is turned to a 2-layer DLHC of GaAs (Figure 5d), the degenerate VBM(CBM) of DLHC in Figure 5c split again to band edge states with different parities. Due to the energy level repulsion of the valence and conduction bands, the gap is closed and the bands across the Fermi energy are inverted. In this way, the 2-layer DLHC GaAs becomes a topological insulator with Z_2_ = 1. Similar topological properties are also found in other 2-layer DLHC materials like InSb, InAs, GaSb, HgTe, and AlSb [30,44]. Moreover, the topological property can be controlled by engineering 2D van der Waals heterostructures with ferroelectric materials, which will be discussed in Section 2.7.

### 2.5. Optical Properties

The reported optical properties of DLHC mainly focus on the light absorptions predicted by first-principles calculations. The traditional III–V semiconductors have traditionally been used for luminescence, so most attention is paid to III–V DLHCs including AlAs [30], AlSb [33,45], InAs [35], InSb [45], AlN, GaN, InN, and AlP [46]. Since the VBM and CBM of DLHC AlAs are of the same parity, despite a direct gap of 2.0 eV at Γ, the optical transition only becomes significant when it approaches 3.14 eV (corresponding to a VBM → CBM + 1 transition) [30]. The absorption spectra of other DLHCs range from infrared light to ultraviolet light. The energy of the first absorption peak of AlSb is smaller than the direct band gap that exhibits an excitonic instability and spontaneous exciton condensation which is the role played by the spin-orbit coupling (SOC) [33]. Table 2 compares the optical properties of AlSb and InSb, yielding the result that the absorption spectrum of the DLHC AlSb is slightly shifted towards higher energy than the corresponding spectrum of the DLHC InSb because of their semiconducting (AlSb) and metallic (InSb) character, respectively [45]. AlN, GaN, InN, AlP, and AlAs are indirect semiconductors that can absorb ultraviolet radiation. Among them, DLHC InN and AlAs may have a stronger ability to absorb visible light [46]. Interestingly, a recent work [47] predicts that DLHC CuI is a p-type semiconductor with a small effective hole mass, while it is also transparent in the visible and near ultraviolet spectral range due to the relatively large band gap (E_g_ ≈ 3.3 eV by HSE calculation). A report proposes that, due to the ultraviolet absorption at ∼350 nm, I–VII DLHC (including CuBr, CuI, AgBr, and AgI) could be used for Ce^3+^-Yb^3+^ co-doped Y_3_Al_5_O_12_ (YAG) quantum cutting which may benefit the Si-based solar cell [39]. Since the study of DLHC materials is still ongoing, more explorations on optoelectronic applications are desired.

### 2.6. Defect and Absorption of Metal Atoms

Defect physics in semiconductors is the basis for regulating their electrical conductivity properties and thus is one of the cornerstones of electronic devices [48]. Recently, the defects of vacancy and substitutional doping in DLHC AlSb are systematically investigated [49]. Note that a previous experiment has confirmed the successful fabrication of DLHC AlSb [34]. First-principles calculations using the HSE method reveal intrinsic DLHC AlSb has a direct band gap at Γ point. Figure 6 shows the local structures of vacancy defects (V_Al_, V_Sb_, and V_Al_ + V_Sb_), substitutional doping defects on the Al site X_Al_ (X = Li, Mg, Si, Ge, Sb), and the Sb site X_Sb_ (X = Li, Mg, Al, C, Si, O, S). The authors claim that the stability of these defect structures is confirmed by MD simulations at 400 K [49]. The table (Figure 6p) shows the charge transfer (Δq) in which a positive (negative) value means the electrons transfer from the substitutional defect atom (DLHC) to DLHC (substitutional defect atom). According to the table, the C_Sb_ may be a good acceptor because it has the largest transferring charge and smaller formation energy. As for donor defects, Si_Al_ has the smallest formation energy and Mg_Al_ has the larger transferring charge. In addition, Mg_Sb_ shows a ferromagnetic property. However, the analyses on charge transfers alone may not be enough to fully identify donors or acceptors of defects. More investigations of defect behaviors in DLHC materials are needed. Especially, transition level calculations should be required to evaluate the ionization energy of various defects in these new 2D materials [50,51,52].

In addition, the absorption of alkali metal (AM) ions can also change the electrical property of DLHC materials. A theoretical study proposes that DLHC AlP can be a potential anode material for AM-ion batteries according to its suitable chemical absorption ability, good electronic conductance, and favor for ion migration [53]. According to absorption energy, the hollow site on the top of the honeycomb is the most favorable site for AM-ions. The system becomes metallic after the adsorption which may benefit electronic conductivity. DLHC AlP is also proposed to be a choice of a cathode active catalyst, which can lead to O_2_ dissociation at a specific site with a low energy barrier [54].

### 2.7. Multilayer and Heterojunction

Compared with monolayer 2D materials, multilayer or heterojunction 2D materials often have new properties and thus show potential new applications. It is demonstrated that DLHC multilayer structures can be stable within a specific thickness because they have lower total energies than other 2D counterparts including the slab truncated from their bulk materials [44]. For example, the maximum number of layers for DLHC AlSb and DLHC ZnTe is 5. However, beyond 5 layers, the DLHC multilayer is no more a stable structure. This is because the surface-volume ratio becomes smaller with the increase in slab thickness. The surface of the bulk-truncated slab is unstable compared with the vdW surface of DLHC. But the chemical bonds inside the slab are stronger than the vdW interactions in multilayer DLHC. Therefore, the bulk-truncated slab gradually becomes stable with the increase of slab thickness owing to the reduction of the surface-volume ratio.

In addition, heterojunctions of DLHC and other 2D materials are also predicted to have novel properties. For example, the transition-metal substrate can yield large-area high-quality graphene but disturb the Dirac cone of graphene. Intercalating two-dimensional semiconductors into them, such as using DLHC structures of CuI, AgI, AlAs, MgSe, and ZnS, can decouple metal substrate and graphene, and thus recover the Dirac states [55,56]. It also provides a method to tune carrier concentrations of graphene with 2D materials. Recently, Mamiyev et al., have shown the ordered Sn-interface structures, such as the new triangular lattice of Sn, can be achieved by intercalation [57]. Moreover, in [58], spatially well-defined graphene p-n junctions can be engineered at the nanoscale by H-intercalation. It is worth mentioning that bilayer graphene can also be obtained by intercalation. As such, intercalation should be a potential approach to obtaining new 2D materials and also controlling their electronic properties. In fact, the present reports of the growth of DLHC materials usually employ an extra buffer layer above a substrate. Therefore, intercalation should be also important for the growth of DLHC materials. 

The vdW superlattice composed of DLHC GaAs and graphene or h-BN also shows the band inversion of DLHC GaAs overlapping the Dirac cone of the graphene [59]. A more interesting example is that the heterojunction composed of 2D α-In_2_Se_3_ and DLHC CuI can be considered a ferroelectric quantum spin Hall insulator [60]. Figure 7a shows the atomic structure of 2D α-In_2_Se_3_ and DLHC CuI. The VBM of isolated DLHC CuI and the CBM of isolated 2D α-In_2_Se_3_ overlap with an energy difference of −0.54 eV ($i.e.,\;E_{C({\mathrm{In}}_{2}{\mathrm{Se}}_{3})}-E_{V\left(\mathrm{CuI}\right)}$), as shown in Figure 7b. However, 2D α-In_2_Se_3_ is a ferroelectric material with a potential energy difference of Δϕ = 1.12 eV as is shown in Figure 7c. As such, the property of the heterojunction composed of 2D α-In_2_Se_3_ and DLHC CuI depends on the stacking sequence. Figure 7d illustrates the heterojunction energy bands change from a trivial semiconductor to a quantum spin Hall insulator with a non-trivial Z_2_ = 1 topological invariant when the polarization is switching from pointing toward the DLHC CuI layer to the opposite direction. Their electronic band structures are also shown in Figure 7e,f. In addition, the SOC effect has a very strong influence on the DLHC CuI energy band and opens a topological gap of 50 meV, as shown in Figure 7f. More exotic properties of DLHC-related heterojunctions can be expected.

## 3. Conclusion and Future Directions

In summary, this work reviews the structures, properties, potential applications, and bonding mechanisms of the emerging new type of 2D DLHC materials. Theoretical investigations demonstrate that the DLHC configuration can exist stably at low formation energy when the thickness of the III–V, II–VI, and I–VII bulk materials is reduced to two atomic layers. Some candidates of DLHC materials are also successfully verified by recent epitaxy-growth experiments. The properties of an excitonic insulator, a topological insulator, optical properties, chemical properties, defect engineering, multilayer, and heterojunctions are discussed in detail. The present results suggest DLHC not only can be a promising platform to investigate topological physics but also shows potential in optoelectronic devices such as detectors. Therefore, DLHC can be a unique structure in achieving new 2D materials with interesting properties from traditional semiconductors. 

Since DLHC is still a newly ongoing topic, we propose several outlooks: (i) More experimental validations are needed. (ii) The I–VII DLHCs may be an important candidate because I–VII 2D materials are rarely reported before. (iii) The defect engineering and the ionization energy of the defects in DLHC are required to be further analyzed for their applications on electronic devices. More investigations of defect behaviors in different conditions, such as temperature, strain, and heterojunctions, are also necessary. (iv) More quantum phenomena like excitonic/topological insulators and their applications in electronic, optoelectronic, and spintronic devices are worth looking forward to. (v) It is also interesting to explore more DLHC materials beyond semiconductors, for example, metallic DLHCs or even superconducting DLHCs. We hope this review can be helpful to timely understand this rising DLHC 2D material and accelerate the development of DLHC-related nanomaterials for future applications.

## Figures and Tables

**Figure 1 materials-15-07715-f001:**
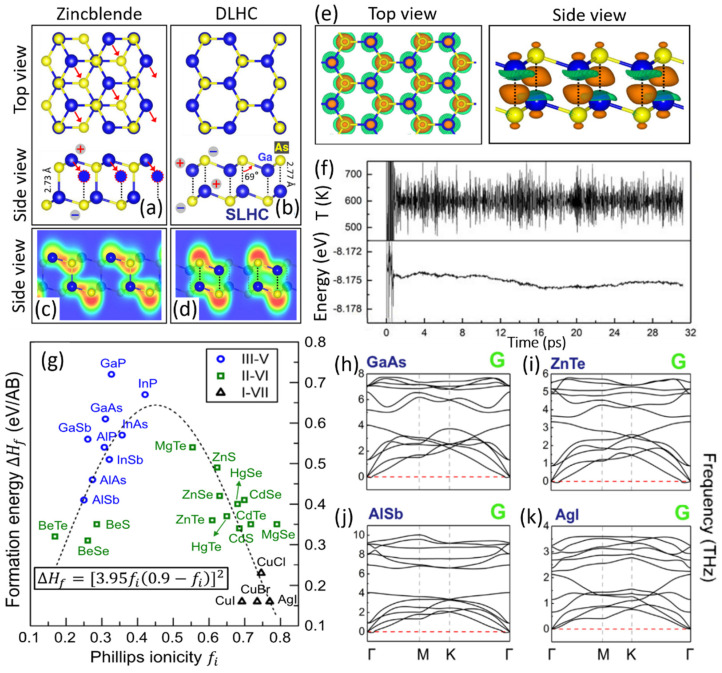
Top and side views of (**a**) a bilayer-thick monolayer truncated bulk GaAs and (**b**) a DLHC GaAs. The red arrows in (**a**) indicate atomic displacements that turn truncated bulk to DLHC. Electron localization functions of (**c**) the truncated bulk GaAs and (**d**) the DLHC GaAs with contour values ranging from 0 (blue) to 0.8 (red). (**e**) Charge density difference (CDD) between DLHC and SLHCs for GaAs with the brown being positive and green being negative. (**f**) The temperature and energy in an MD simulation of GaAs DLHC structure at T = 600 K for 31 ps. (**g**) Formation energy of different DLHC materials as a function of Phillips ionicity. The dotted line is a two-parameter least-squares fit of the data. (**h**–**k**) Phonon spectra of several selected DLHC structures, including group III–V (such as GaAs and AlSb), group II–VI (such as ZnTe), and group I–VII (such as AgI). Reproduced with permission [30]. Copyright 2018, American Physical Society.

**Figure 2 materials-15-07715-f002:**
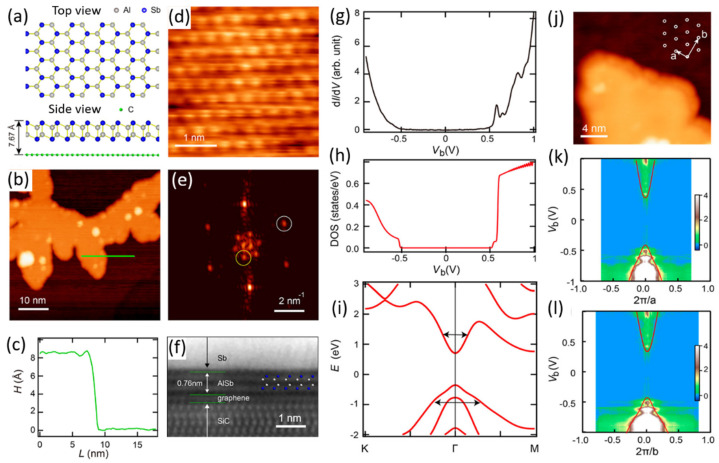
(**a**) Top and side views of DLHC AlSb model on graphene. (**b**) STM topography (V_b_ = 3 V, I_t_ = 5 pA) and (**c**) apparent height along the green line in (**b**). (**d**) STM image with atomic resolution (V_b_ = 0.7 V, I_t_ = 30 pA) of AlSb and (**e**) fast Fourier transformation of (**d**). (**f**) STEM image of the cross-sectional view of the AlSb film, whose thickness agrees well with the theoretical calculation in (**a**). (**g**) Averaged tunneling spectrum of DLHC AlSb (V_b_ = 1 V, I_t_ = 100 pA, V_mod_ = 20 mV). (**h**) Density of states and (**i**) band structures of DLHC AlSb, obtained by HSE@G_0_W_0_ approach. The arrows indicate the flatness of conduction and valence bands. (**j**) STM topography (V_b_ = 3 V, I_t_ = 5 pA) of a DLHC AlSb island. Insert is a schematic of the top layer Sb atoms with unit vectors marked. Energy dispersion relation along the (**k**) Γ-K and (**l**) Γ-M directions. Reproduced with permission [34]. Copyright 2021, American Chemical Society.

**Figure 3 materials-15-07715-f003:**
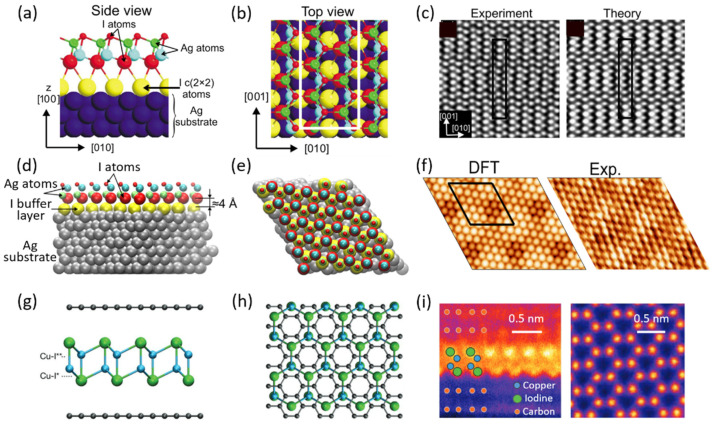
(**a**) Side and (**b**) top views of the sandwich-like AgI film on Ag(100) surface. (**c**) Theoretical and experimental simulated STM images of the AgI with a distorted DLHC structure (U_s_ = −260 mV). Reproduced with permission [36]. Copyright 2009, American Physical Society. (**d**) Side and (**e**) top views of AgI film on Ag(111) surface. (**f**) Comparison of theoretical and experimental STM images of the surface (Us = −500 mV). Reproduced with permission [37]. Copyright 2022, AIP Publishing. (**g**) Side and (**h**) top views of CuI sandwiched by graphene layers. The “Cu–I*” and “Cu–I**” mean the in-plane and out-of-plane bonds, respectively. (**i**) Cross section STEM and top high-angle annular dark-field (HAADF) images of the synthesized 2D CuI. Reproduced with permission [38]. Copyright 2022, John Wiley and Sons.

**Figure 4 materials-15-07715-f004:**
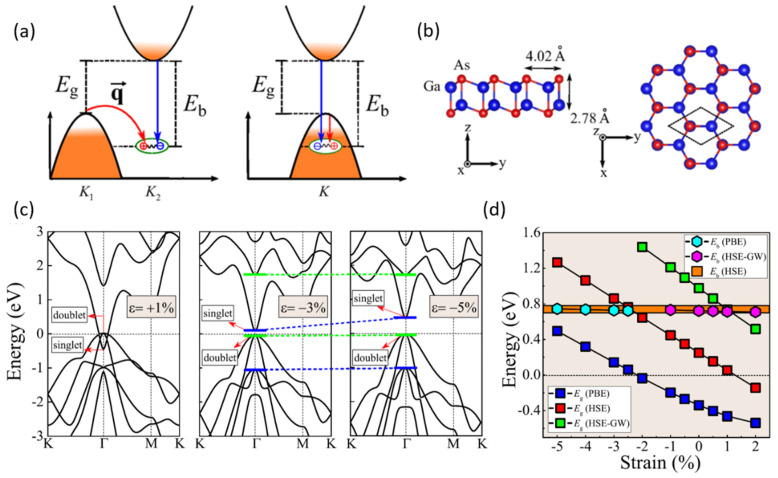
(**a**) A schematic illustration of excitonic instability in indirect- and direct-gap materials. (**b**) Side and top views of DLHC GaAs. (**c**) Band structures by PBE of the 2D GaAs under different strains ε. The opposite parities are marked in short green and blue bars, respectively. (**d**) Strain dependence of E_g_ and E_b_, calculated by PBE, HSE, and GW, demonstrating the decoupling between E_g_ and E_b_. Reproduced with permission [42]. Copyright 2018, American Physical Society.

**Figure 5 materials-15-07715-f005:**
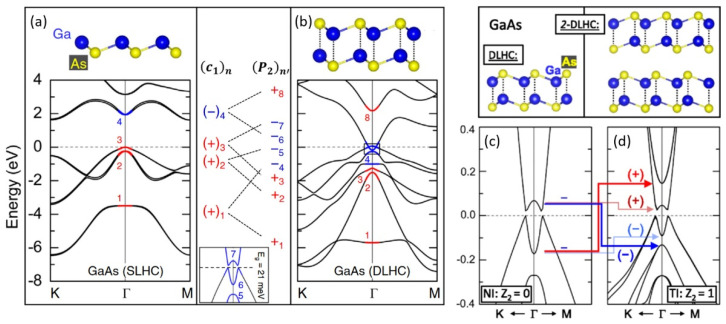
Atomic and band structures of (**a**) SLHC and (**b**) DLHC GaAs by HSE calculations. Inset highlights the band inversion of (**b**). The red means even parity and blue means odd. Band structure of (**c**) DLHC and (**d**) 2-layer DLHC GaAs. Reproduced with permission [30]. Copyright 2018, American Physical Society.

**Figure 6 materials-15-07715-f006:**
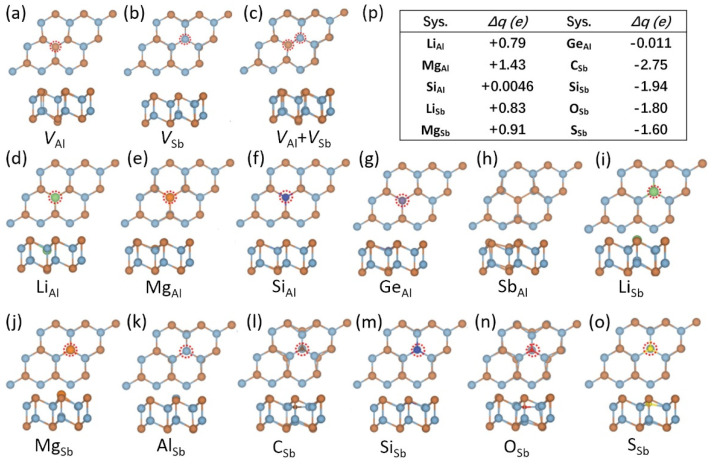
Atomic structures of DLHC AlSb with vacancy and substitutional defects: (**a**) V_Al_, (**b**) V_Sb_, (**c**) V_Al_ + V_Sb_, (**d**) Li_Al_, (**e**) Mg_Al_, (**f**) Si_Al_, (**g**) Ge_Al_, (**h**) Sb_Al_, (**i**) Li_Sb_, (**j**) Mg_Sb_, (**k**) Al_Sb_, (**l**) C_Sb_, (**m**) Si_Sb_, (**n**) O_Sb_ and (**o**) S_Sb_. (**p**) Charge transfer (Δq) of these defects, where the positive (negative) values indicate defect tends to lose (gain) charge. Reproduced with permission [49]. Copyright 2021, IOP Publishing.

**Figure 7 materials-15-07715-f007:**
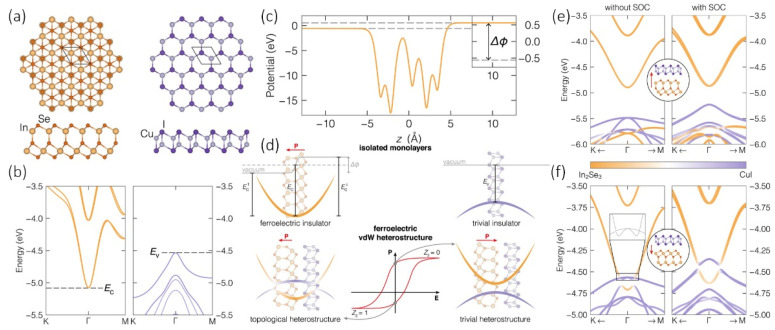
(**a**) Top and side views of isolated α-In_2_Se_3_ (left) and DLHC CuI (right) monolayers. (**b**) Electronic band structures obtained by DFT calculations. (**c**) The electrostatic potential energy Δφ of monolayer α-In_2_Se_3_ along the vertical coordinate z. (**d**) The principles of ferroelectric control of the topological property in the α-In_2_Se_3_/DLHC CuI heterostructure. The band structures with the ferroelectric polarization (**e**) upwards and (**f**) downwards. The left and right panels show the band structures obtained without and with the effect of spin-orbit coupling (SOC), respectively. Reproduced with permission [60]. Copyright 2022, Springer Nature.

**Table 1 materials-15-07715-t001:** Experimental conditions of the growth of different DLHC materials.

Ref.	Material	Substrate	Experimental Method	Temperature
[34]	AlSb	Graphene-covered SiC (0001)	molecular beam epitaxy	substrate optimal temperature T_s_ = 170 °C
[36]	AgI	I buffer layer covered Ag (100)	Chemisorption of monolayer iodine atoms under ultra-high vacuum (UHV) conditions	Room temperature
[37]	AgI	I buffer layer covered Ag (111)	Chemisorption of molecular iodine atoms under ultra-high vacuum (UHV) conditions	Room temperature
[38]	CuI/AgI	Two Graphene layers on Graphene oxide	Synthesized directly between graphene encapsulation by wet-chemical process	Room temperature

**Table 2 materials-15-07715-t002:** Calculated optical absorption properties of DLHC materials. “I” and “D” are for indirect and direct band-gap semiconductors, respectively.

Ref.	Materials	Band Gap	Method	Position of Absorption Peak	Main Absorption Regions
[30]	AlAs	2.0 eV (I)	HSE + SOC + D3	3.14 eV (First)	Excitonic insulator with same parity CBM and VBM
[33]	AlSb	1.35 eV (D)	G_0_W_0_	0.82 eV (First)	exciton adsorption, with a binding energy of 0.53 eV.
		0.74 eV (D)	G_0_W_0_ + SOC	0.65 eV (First)	just 0.09 eV lower than the quasiparticle gap
[35]	InAs	0.24 eV (D)	HSE06	Nearly 1.1 eV (First)	visible, ultraviolet
[45]	AlSb	0.08 eV (D)	GGA + SOC	0.75 eV for the first peak and 5 eV for the main peak	visible, infrared spectra, and activated in the ultraviolet region.
		0.9 eV (D)	HSE		
		0.7 eV (D)	HSE + SOC		
	InSb	0.10 eV (D)	GGA + SOC		
		0.06 eV (D)	HSE		
		0.09 eV (D)	HSE + SOC		
[46]	AlN	3.54 eV (I)	PBE	above 4.0 eV (First)	ultraviolet
		4.71 eV (I)	HSE06		
	GaN	1.78 eV (I)	PBE	above 3 eV (First)	ultraviolet
		2.99 eV (I)	HSE06		
	InN	0.18 eV (I)	PBE	below 1.0 eV (First)	visible, ultraviolet
		1.16 eV (I)	HSE06		
	AlP	1.68 eV (I)	PBE	4.2 eV (main peak, xx directions) and 6.0 eV (main peak, zz directions)	ultraviolet
		2.41 eV (I)	HSE06		
	AlAs	1.28 eV (I)	PBE	3.8 eV (main peak, xx directions) and 5.8 eV (main peak, zz directions)	visible, ultraviolet
		1.93 eV (I)	HSE06		
[39]	CuBr	3.198 eV (D)	HSE06	About 8 eV (main peak)	ultraviolet
	CuI	3.117 eV (D)	HSE06		
	AgBr	3.358 eV (D)	HSE06		
	AgI	3.163 eV (D)	HSE06		
[47]	CuI	3.7 eV (D)	PBE0 + SOC	About 7 eV	transparency in the visible and near ultraviolet spectral range
		1.8 eV (D)	PBE + SOC		

## Data Availability

Additional data can be obtained upon reasonable request from the corresponding author.

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
