# Peer review of "Recent Progress in Double-Layer Honeycomb Structure: A New Type of Two-Dimensional Material"

_materials, 2022, doi:10.3390/ma15217715_

Round 1

Reviewer 1 Report

The review by Ming-Yu Ma et al. aims discussion on the layered hexagonal lattices, which have been investigated in the last couple of years. The field of research is interesting. However, the very limited number of publications, also seen from the extremely short reference list here, shows that it is not a fast-growing field. Taking this into account, I have the following suggestions, which are required for further processing of this review paper.

1. The reference list and discussions should be extended.

2. The title doesn't mention anything about metals or semiconductors; however, the authors discuss mainly semiconductors. Therefore, I think adding some new aspects of metallic DLHC would already extend the discussions.

3. Some of the sections, such as 5, 6, etc., are extremely short and don't contain significant discussion. 

4. In section 8, I suggest the authors mention the intercalation of metallic species as well. For example, next to this sentence, "The intercalating of two-dimensional semiconductors into them, such as DLHC structures of CuI, AgI, AlAs, MgSe, and ZnS, can decouple the metal and the graphene, thus recover the Dirac states of the graphene, and also provide a method to tune carrier concentrations of graphene with 2D materials [30,31]. For this, I suggest citing the following papers. Indeed, these works show that Gr can effectively be decoupled from the substrate by intercalation (even using H), and especially in ref 1, the new triangular lattice of Sn could be achieved. Moreover, in ref 2, p-n junction in GNRs could be achieved by H intercalation. I would mention that when intercalating the monolayer graphene, it ends up with the bilayer graphene. This indeed opens a new avenue for the DLHC formation by confined epitaxy and/or intercalation.

Ref 1. https://doi.org/10.1016/j.surfin.2022.102304

Ref 2. https://doi.org/10.1002/adfm.202109839

5. Some of the very important discussions are cut in half, which should not be the case. An exemplary quote from section 8: "For example, the maximum number of the layers for DLHC AlSb and DLHC 271 ZnTe is 5. However, beyond 5 layers, DLHC multilayer is no more the most stable structure." Indeed, such half statements without any description of the discussed phenomenon don't add any value to the field.

6. In general, the manuscript in its present form seems more like a short summary of references, with a lack of discussions. Moreover, the text contains several typos, in addition to a lack of flow, which make reading hard. Therefore, I suggest the authors read their manuscript critically and extend it with discussions. I hope my comments will be helpful in this regard. I will be willing to review the renewed manuscript.  

Author Response

Dear colleagues of Materials,

We sincerely thank you for your efforts on the work. We also appreciate referees for their positive evaluations and professional suggestions on our review. Below we reply point-by-point to all the concerns of the referees and make corresponding revisions in the manuscript.

Thank you again and best wishes,

Prof. Nian-Ke Chen and Prof. Xian-Bin Li

Reviewer 2 Report

This review article about double layer honeycomb structure is very well written and included recent research findings from theory to experiment. I think this review will be a very good introductory reference to this topic.  

Author Response

Dear colleagues of Materials,

We sincerely thank you for your efforts on the work. We also appreciate the referee for the very positive evaluation that “This review will be a very good introductory reference to this topic.” and other appreciations of our articles.

Thank you again for your fully supporting the publication and best wishes,

Prof. Nian-Ke Chen and Prof. Xian-Bin Li

Reviewer 3 Report

REVIEW

on article

Double-Layer Honeycomb Structure:

A New Type of Two-Dimensional Materials

Ming-Yu Ma, Dong Han, Nian-Ke Chen, Dan Wang and Xian-Bin L

SUMMARY

The article submitted for review is devoted to a topical issue. 2D materials are among the most studied nanomaterials in recent decades, that is, the study is relevant and modern, which fully meets the requirements of the research agenda. In terms of depth of study, the authors note that they have applied structural studies and experimental validation, as well as applied high analytical approaches to research and carried out a very in-depth review, which made it possible to assess the current state of the issue of two-layer honeycomb structures of new types of two-dimensional materials. Thus, their review is useful and can help the scientific and engineering community to better understand DLHS materials and inspire the world's industry to use these materials. With all the advantages of the article, this review has a number of shortcomings. They should be corrected before publishing this article in Materials journal.

COMMENTS

1.    The Abstract presented by the authors does not perform the options that it should perform. In particular, the authors are overly carried away by the description of relevance. The first 3 sentences testify to one thing and that 2D materials are widely studied, popular, relevant. Research problems arise only at the 4th sentence in the abstract. Thus, there is some bias, due to which the Abstract is hard to perceive and hard to understand the content of the article.

2.    In addition, it is not clear what scientific problem the authors were solving. As a rule, any literature review contains an analytical component and is designed to eliminate existing scientific deficits. However, the authors abruptly move after the relevance and insufficiently formulated problem to the methodology of the study, describe what they did, and at the end talk about the practical significance of this study. However, the abstract does not show what their scientific result is. Therefore, the authors should rework the Abstract.

3.    The authors used 4 key phrases. This is not enough to make your research accessible and visible in search engines. In addition, the phrase Double-Layer Honeycomb Structure is not very well chosen, since it contains 4 words in a row. You should probably enter more fractional keywords, as this narrows the search very much.

4.    The Introduction presented by the authors is very short in order to reflect the scientific novelty and the problem raised, as well as the goals and tasks of the study. The authors give a small overview of only 11 sources and, on the basis of this, set the goal and tasks. Methodologically, this is incorrect, because in this case the depth of the study is not clear. In addition, on line 43, the presence of links to 3 sources at once is striking, that is, some of the sources are considered only superficially, therefore it is not clear how deeply the analysis was carried out by the authors.

5.    In Section 2, Figure 1 is saturated, with most of the characters on it being almost unreadable, which is unacceptable. It is necessary to divide it into different figures, as is done in the figure caption from Figure 1a to Figure 1k. Thus, the authors must work with the graphical representation. The same remark applies to Figure 2.

6.    Figure 3 is made in a higher quality, however, there are also unreadable characters on it. Needs to be improved.

7.    The authors got carried away with excessive compression and placement of a large number of diverse graphic elements on the same figures. The reviewer is not entirely clear why this was done. According to the reviewer, this makes the content of the article heavier and makes some points in the article incomprehensible.

8.    It is not clear on what basis the division into sections was carried out. Perhaps the authors should have paid attention to the methodology of the study, where it was worth clearly identifying the scientific problem, goals, tasks, methods and likely approaches that the authors used. Why did they break their research according to such criteria, was the classification and grouping of those new knowledge or analyzed studies of other authors carried out on the topic that the authors revealed?

9.    The lack of a proper discussion of the results is noteworthy. Probably, the authors should have provided comparative tables, graphs and other quantitative measures of comparing the results of various authors with each other.

10.  The scientific value of the study is not clear. At the moment, it looks like a collected narrative of related studies and looks like a collection, and it should contain in-depth analysis, which is placed in the discussion section.

11.  The conclusions are also made somewhat chaotically and need some refinement in terms of a clear formulation of the scientific result and determination of further research prospects.

12.  The authors declare the specified material as a review. However, only 33 sources have been analyzed, with the authors contradicting themselves in that they themselves state that 2D materials are among the most studied materials in recent times. Thus, the analyzed 33 sources cause some confusion. At a minimum, the authors had to analyze about 100 sources, which is easily confirmed by searching through systems, such as MDPI. Thus, the authors performed only an initial, approximate study, a vector was set. But the study needs serious improvements and in this form it cannot be published in Materials. The article should be enlarged several times in terms of the analyzed sources, the analytical component should be supplemented. In addition, authors should refer to the checklist on the MDPI platform for a clearer understanding of the structure of a review article so that it meets all the established requirements.

Author Response

Dear colleague of Materials,

We sincerely thank you for your efforts on the work. We also appreciate referee for their positive evaluations and professional suggestions on our review. Below we reply point-by-point to all the concerns of the referees and make corresponding revisions in the manuscript.

Thank you again and best wishes,

Prof. Nian-Ke Chen and Prof. Xian-Bin Li

Round 2

Reviewer 1 Report

The authors have adequately addressed my comments and improved the manuscript. The paper can be accepted for publication in the present form.

Reviewer 3 Report

All my comments were taken into account and the manuscript looks much better. 

I recommend this Review for publishing.